# The Mechanical Properties of Aluminum Metal Matrix Composites Processed by High-Pressure Torsion and Powder Metallurgy

**DOI:** 10.3390/ma15248827

**Published:** 2022-12-10

**Authors:** Mohamed Ibrahim Abd El Aal, Hossam Hemdan El-Fahhar, Abdelkarim Yousif Mohamed, Elshafey Ahmed Gadallah

**Affiliations:** 1Department of Mechanical Engineering, College of Engineering in Wadi Alddawaser, Prince Sattam Bin Abdulaziz University, Wadi Addawaser 18734, Saudi Arabia; 2Mechanical Design & Production Department, Faculty of Engineering, Zagazig University, Zagazig 44519, Egypt; 3Mechanical Production Department, Faculty of Technology & Education, Suez University, Suez 43527, Egypt

**Keywords:** metal matrix composites (MMCs), high-pressure torsion (HPT), microstructure evolution, mechanical properties

## Abstract

Al-Al_2_O_3_ and SiC metal matrix composites (MMCs) samples with different volume fractions up to 20% were produced by high-pressure torsion (HPT) using 10 GPa for 30 revolutions of Al-Al_2_O_3_, and SiC and powder metallurgy (PM). The effect of the processing method of micro-size Al MMCs on the density, microstructure evolution, mechanical properties, and tensile fracture mode was thoroughly investigated. HPT processing produces fully dense samples relative to those produced using powder metallurgy (PM). The HPT of the Al MMCs reduces the Al matrix grain size and fragmentation of the reinforcement particles. The Al matrix average grain size decreased to 0.39, 0.23, and 0.2 µm after the HPT processing of Al, Al-20% Al_2_O_3_, and SiC samples. Moreover, Al_2_O_3_ and SiC particle sizes decreased from 31.7 and 25.5 µm to 0.15 and 0.13 µm with a 99.5% decrease. The production of ultrafine grain (UFG) composite samples effectively improves the microhardness and tensile strength of the Al and Al MMCs by 31–88% and 10–110% over those of the PM-processed samples. The good bonding between the Al matrix and reinforcement particles noted in the HPTed Al MMCs increases the strength relative to the PM samples. The tensile fracture surface morphology results confirm the tensile properties results.

## 1. Introduction

The need for new materials with high properties and lightweight, which reduce the final product’s power consumption, can be considered one of the essential branches of scientific research in the last 20 years. Therefore, the need to provide new materials with high properties, such as the (MMCs), is still required. The increase in the (MMCs) contribution to different industries makes them the pioneer selection in the military, aerospace, automotive, and sports goods industries [1,2]. In the last 30 years, the improvement of (MMCs) properties motivated material science research [1,2]. The improvement of the MMCs’ isotropic mechanical and physical properties makes them a superior candidate in different applications over other materials [3].

MMCs processing methods can be classified into two groups. First, the primary liquid phase matrix processing, such as squeeze casting and spray deposition. Second, the primary solid phase processing, such as diffusion bonding of foils, physical vapor deposition (PVD), and powder metallurgy (PM) processes. The disadvantages of using liquid-phase matrix processing methods, such as insufficient wetting of the ceramic particle, turn the rudder in the direction of using the primary solid-phase methods, especially PM [4,5]. In recent years, the fabrication of ultrafine (UFG) and nanomaterials MMCs with superior properties has been the trend in material science society research. The UFG and nanomaterials MMCs can be produced by consolidating UFG or nano-powders using down-to-up techniques. Unfortunately, this method includes many defects, such as inclusions, contamination, and a high percentage of voids. Therefore, decreasing the matrix grain and reinforcement particle sizes using the up-to-down techniques was the best selection to overcome the problems in the down-to-up technique.

One of the essential methods of up-to-down techniques is severe plastic deformation (SPD). The possibility of fabricating different UFG and nano MMCs using (SPD) has recently attracted attention due to the specific properties of materials with such a structure, which are very promising for different purposes and applications in engineering. The advantage of various SPD techniques aiming at grain refinement and strength enhancement is the possibility of performing large plastic deformations without changing the billet shape due to local deformation. High strength and material integrity retain, which is impossible to achieve by applying other techniques for processing bulk materials with UFG or a nanostructure (e.g., by powder consolidation).

Different micro, UFG, and nano MMCs samples were fabricated by equal-channel angular pressing (ECAP) and high-pressure torsion (HPT) successfully using powder and chip matrix [6,7,8,9,10,11,12,13,14,15,16,17]. Moreover, the SPD processing of MMCs and intermetallics materials improve their mechanical properties obviously [18,19]. Interestingly, SPD processing of the MMCs effectively overcame many disadvantages of the PM, such as porosity and low mechanical properties, as previously noted [16].

The observations obtained in the previous works about processing the MMCs by SPD processes can be summarized below. First, the SPD processing effectively refined the MMCs matrix grains [6,7,8,9,10,11,12,13,14,15,16,17,20,21]. Second, an apparent agglomeration of reinforcement particles was evident during the SPD processing of nano-reinforcement MMCs combined with ball milling [9,10,11,12]. Third, the SPD processes have a noticeable influence on the re-distribution and fragmentation of reinforcement [6,7,8,13,14,15,16,17,20,21]. However, there are conflicts in the previous results about the influence of SPD processing on the fragmentation of the reinforcement particles. However, most previous works support the effectiveness of SPD processing in the fragmentation of ceramic particles [7,8,14,15,16,17].

The SPD processing of MMCs indicates the effectiveness of starting with micro-size reinforcement [7,8,14,15,16,17,20,21]. SPD of micro-size reinforced MMCs capable of producing MMCs samples with homogenized distributed reinforcement particles, a porosity-free refined matrix, and fragmented reinforcement [7,8,14,15,16,17,20,21].

However, there is an apparent need for further research on SPD-processed Al MMCs with different volume fractions. Therefore new investigations need to be carried out to overcome the shortage previously noted in the limited results available about the correlation between microstructure evolution during the HPT of micro-size Al MMCs and the enhancement of the hardness and tensile properties. Furthermore, the need to compare the advantages of using SPD processing of the Al MMCs relative to traditional methods such as PM is still required.

The present work was undertaken to reach the following aims:Investigate the feasibility of fabricating HPTed Al MMCs reinforced with SiC and Al_2_O_3_ with high relative density;Investigate the influence of using HPT on the Al matrix grain refinement and reinforcement fragmentation of the Al MMCs;Investigate the effect of the fabrication method on the microhardness and tensile properties of the Al MMCs;Investigate the effect of the HPT processing on the fracture mode and morphology of the tensile fracture samples of the Al MMCs;Introduce a complete comparison between the PM and HPT-processed Al MMCs to select the most effective fabrication method for producing Al MMCs.

## 2. Materials and Methods

Al, Al_2_O_3_, and SiC powder with a purity of 99% and particles size of 78.8, 31.6, and 25.5 µm were used to fabricate a 10 mm diameter Al, Al-10%, and 20% of Al_2_O_3_, and SiC samples by PM. PM was performed by pressing under 600 MPa and then sintered at 600 °C for 6 h. Furthermore, 10 mm diameter cold compacted samples were HPTed under 10 GPa, for 30 revolutions at room temperature (RT) and a rotational speed of 1 rpm. More details about the PM and HPT steps are indicated in the flow chart in Figure 1.

The samples were prepared to measure the experimental density using METTLER TOLEDO’s XS204 device. Moreover, the values of the theoretical densities were obtained through the rule of the mixture [1]. Therefore, the relative density (which indicates the degree of the densification and the void% content) was calculated based on equations previously used [16].

The microstructure was thoroughly investigated for all samples using different microscopes. The PM and HPTed samples were ground, then polished and etched using diamond paste suspensions and Keller’s reagent. OLYMPUS BX51M optical (OP) and a scanning electron microscope (SEM; model XL30SFEG, Philips, Tokyo, Japan) microscopes were used to investigate the matrix grains and reinforcement particle sizes and distribution of the PM and HPTed samples.

The Al matrix grain refinement tracing was performed by CS-corrected field emission transmission electron microscopy (TEM) (JEOL JEM-2100F, Tokyo, Japan). The HPT samples were ground and polished by a mixture of colloidal silica and ethanol for 2 h to have a shiny surface and final thickness of 100 μm. Disc-shaped specimens with a diameter of 3 mm were punched and then electro-polished. The Al matrix grain size measurement was performed by the line intercept method [22,23].

Mechanical testing was performed to assess the influence of the manufacturing method on the mechanical properties of the Al and Al MMCs. The microhardness was measured along the sample’s diameter. A Mitutoyo microhardness tester equipped with a Vickers indenter was used under an applied load of 100 gf and a dwell time of 15 s. A total of 40 measurements were obtained with a space of 0.25 mm between every two measurements. Each sample microhardness’s average value and the inhomogeneity index (by the equation previously used [24]) were calculated.

The tensile properties at RT were thoroughly investigated. Micro-tensile samples of a 1.5 mm gauge length machined using an electrical discharging machine. A Micro-tensile test was carried out using a UNITECH Micro load machine under a strain rate of 10^−3^ s^−1^ using a special grip [16]. The fracture surface morphology and fracture mode were also studied using SEM. The flow chart of the experimental testing and the positions of the OP, SEM, and TEM observations and tensile test samples are shown in Figure 2.

## 3. Results and Discussion

### 3.1. Density Results

Density considers the indication that reflects the degree of densification and voids% of the processed powder samples. The experimental densities of the different Al and Al MMCs samples are shown in Figure 3. Moreover, based on the theoretical densities of the Al, Al-20% Al_2_O_3_, and SiC samples of 2.7, 3.0, and 2.8 g/cm^3^, respectively, and by using the rule of mixture [1,16], the relative densities were calculated and presented (Figure 3).

The HPTed samples have higher densities than those of the PM samples. Therefore, HPT processing can produce more densified samples than those processed by PM. This observation was also confirmed by calculating the void% from Equation (1).
(1)$$\mathrm{Void}\%=\frac{\rho_{\mathrm{theo}}-\rho_{\exp}}{\rho_{\mathrm{theo}}}$$
where void%, ρ_theo_, and ρ_exp_ are the void content%, theoretical and experiment density of each sample, respectively. The void% of the PM Al, Al-20% Al_2_O_3_, and SiC samples were 1.9, 4.4, and 3.2%, which were higher than those of 0.1, 0.7, and 0.3% of the HPTed samples. The void% in the HPT was 85–96% relative to that of the PM samples. Therefore, HPT of Al and Al MMCs produced fully dense samples, which are almost free of voids.

Producing approximately fully dense MMCs was also noted after SPD of different MMCs [6,7,9,10,11,12,14,16,17]. The ECAP processing of Al-micro size SiC [6,7] and Al-nano size Al_2_O_3_ [9] produce samples with void content ranging from 1.4 to 2.6% [6], 1.8 to 2.5% [7], and 2.2 to 7.3% [9]. Furthermore, the HPT processing of Al-nano size Al_2_O_3_ [10,11,12], Al-Si-Cu-micro SiC, and Al-micro SiC under 5–20 revolutions and pressure 5–9 GPa revealed void% of 1–2, 0.6, and 1.3–1.7%, respectively. The analysis of the previous results shows that the SPD of the Al MMCs reinforced with micro-size particles [6,7,14,17] results in lower void% than those reinforced with nano-size particles [9,10,11,12]. Therefore, the SPD processing of MMCs reinforced with micro-sized particles produces fully dense samples due to the lower degree of particle agglomeration (as confirmed through microstructure observations in the present work) and observed previously [9,10,11,12]. Interestingly, the void% content of HPTed Al and Al MMCs processed at RT under 30 revolutions and the pressure of 10 GPa was lower than those previously noted [6,7,9,10,11,12,14,17].

### 3.2. Microstructure Observations

#### 3.2.1. Microstructure Observations of the PM Samples

The OP photomicrographs of the PM samples microstructure are shown in (Figure 4 and Figure 5). The PM samples suffer from voids, as indicated by the red arrows. The voids are located in the matrix in the case of the Al sample (Figure 4a). Furthermore, voids were also noted between the reinforcement particles and the interface between the reinforcement particles and the Al matrix (Figure 5). The void% was increased after the formation of the Al MMCs, as observed (Figure 4a and Figure 5). A similar observation of the presence of the voids that consider the essential defect in the PM samples was observed in previous works [16,17,25,26,27]. In addition to the presence of the voids, the MMCs PM samples can also be characterized by the agglomeration of the Al_2_O_3_ and SiC, which increase with increasing the reinforcement volume fraction (Vol%).

The matrix grain size of PM Al, PM Al-10% Al_2_O_3_, PM Al-20% Al_2_O_3_, PM Al-10% SiC, and Al-20% SiC samples was 73.5, 52, 49, 45, and 40 µm, respectively. The increase in the reinforcement Vol% contributes to the Al grain size decrease due to limiting the matrix grain growth. The reinforcement particles still have their initial sizes of 31.6 and 25.5 µm without fracture after the PM processing. Therefore, the PM processing of micro-Al-micro size reinforcement composites produces Al MMCs with relatively high void%, high agglomeration degree, and coarse matrix grain and reinforcement particle sizes.

#### 3.2.2. Microstructure of the HPT Samples

##### TEM Microstructure Observations

Figure 4b and Figure 6 show the TEM photomicrographs of the HPTed samples. The HPT Al sample has an equiaxed grain microstructure with clear grain boundaries and no evidence of voids. The HPT Al sample has a grain size range and an average of 0.47 to 0.06 and 0.39 µm, which is smaller than the 73.5 µm of the PM Al sample. The average grain size of the HPTed Al powder processed up to 30 revolutions and 10 GPa was smaller than those of HPTed Al chip, powder, and solid samples [17,28,29] processed up to 5–30 revolutions under 2.5–9 GPa of 0.54, 0.5, and 0.58 µm, respectively. The high pressure and imposed shear strain can explain the smaller Al sample grain size.

The Al matrix of the HPTed Al MMCs consists of a mixture of equiaxed and elongated UFG with an average grain size of 0.3, 0.23, 0.28, and 0.2 µm in the case of Al-10% Al_2_O_3_, Al-20% Al_2_O_3_, Al-10% SiC, and Al-20% SiC samples, respectively (Figure 6). The Al matrix average grain size of the Al-20% Al_2_O_3_ and SiC was 41 and 51% smaller than the Al sample. Furthermore, the Al matrix average grain size decreased with increasing the reinforcement Vol%. The Al matrix grain size of the HPTed Al-20% Al_2_O_3_ and SiC samples was 23 and 29% smaller than those of Al-10% Al_2_O_3_ and SiC samples.

The grain size decrease with increasing the SiC and Al_2_O_3_ Vol%, and HPT processing is due to the following. The fine Al_2_O _3_ and SiC particles produced from the fragmentation of the initial micro size Al_2_O _3_ and SiC particles (Figure 6 and Figure 7) contribute to the refinement of Al grains. The fine Al_2_O_3_ and SiC particles on the Al grain boundaries prevent the dislocation motion. Consequently, the boundaries play as dislocations accumulation sites, leading to the formation of subgrain and grain boundaries under further straining. Moreover, the dislocation interaction (due to HPT processing that generated a huge number of dislocations) contributes to the evolution of the subgrain into grains with high angles of misorientation (HAGBs).

The decrease in the Al matrix grain size of Al MMCs severely deformed was also previously noted [6,10,11,12,13,14,17,28]. Al matrix grain size decreased from 45 and 35 µm to 8–16 and 0.6 µm after the ECAP processing of Al-5 and 10% SiC and Al-10% Al_2_O_3_ MMCs [6,13]. Moreover, the HPT processing of different micro and nano-reinforced Al MMCs decreases the Al matrix grain size to 0.55–0.06 µm using a pressure of 3–9 GPa and 5–50 revolutions [10,11,12,13,14,17,28]. Therefore, the HPT processing is more effective in producing UFG Al MMCs samples with smaller grain sizes. Moreover, the HPT processing under 30 revolutions and 10 GPa produces UFG Al MMCs samples with smaller Al matrix grain sizes and higher reinforcement Vol% than previously noted [10,11,12,13,14,17,28].

##### SEM Observation of the Al_2_O_3_ and SiC Fragmentation and Distribution

The MMCs Al matrix grain refinement was noted through the TEM observations combined with the fragmentation of the reinforcement (Figure 6). The Al_2_O_3_ average particle sizes decreased from 31.6 µm to 0.26 and 0.15 µm after the HPT of the Al-10 and 20% Al_2_O_3_ (Figure 6). Moreover, The SiC average particle sizes decreased from 25.5 µm to 0.24 and 0.13 µm after the HPT of the Al-10 and 20% SiC samples (Figure 6). The Al_2_O_3_ and SiC particle fragmentation was clear in the TEM dark field photomicrographs, as indicated in (Figure 6c,f).

The reinforcement particle fracture and distribution of the HPTed Al MMCs were also investigated using SEM (Figure 7). The average SiC particle sizes were 0.25 and 0.15 µm, with particle size ranges of 1.21–0.04 and 0.55–0.03 µm after HPT of Al-10 and 20% SiC samples. Moreover, the average Al_2_O_3_ particle sizes were 0.27 and 0.17 µm, with particle size ranges of 1.31–0.05 and 0.89–0.03 µm after HPT of Al-10 and 20% Al_2_O_3_ samples. The results of SEM observation were so close to the TEM one, which proves the accuracy of the obtained results.

A clear fragmentation of the micro size SiC and Al_2_O_3_ particles into ultra-fine UF particles was noted and indicated by arrows (Figure 7c,f). Fragmentation of the reinforcement particles after the HPT is due to the high pressure and imposed strain. Moreover, the low fracture toughness and high brittleness of the SiC and Al_2_O_3_ particles contribute to their fragmentation [30]. The SiC average particle size and range after the HPT processing were lower than those of the Al_2_O_3_ particles (Figure 6 and Figure 7) due to the lower fracture toughness and higher brittleness of the SiC particles.

The fragmentation of the reinforcement particles due to the SPD processing of the MMCs was also previously noted [7,14,15,16,17]. The SiC particle size decreased from 55 to 1 µm after the ECAP of Al-10% SiC [7]. On the other hand, the SiC, W, and Al_2_O_3_ particle sizes decreased from 53, 2–10, 25.5, and 41.4 µm to 11.3, 0.01–0.02, 0.12 and 0.92 µm after the HPT of Al-Si-Cu-5% SiC, W-25% Cu, Cu-10 and 20% SiC, Al-20% SiC, and Al_2_O_3_ [7,14,15,16,17]. Therefore, the SPD of the MMCs effectively fragmented the reinforcement into UF particles. Although limited or even no fragmentation of the reinforcement particles after the SPD of different MMCs was previously noted [6,13,14]. Nevertheless, the limited or even no fragmentation of the reinforcement particles is due to the insufficient imposed strain or pressure applied in the SPD processing. Relative to the apparent imposed strain and pressure used during the HPT, the ECAP processing can be considered insufficient in introducing complete fragmentation of the reinforcement down to UF particles [6,7].

Reaching complete fragmentation of the reinforcement after the HPT of the MMCs can be related to the applied pressure. The HPT applied pressure of 5 GPa or less with values from 3.5 to 5 GPa cannot be considered enough to obtain a complete fragmentation of reinforcement particles to UF or nanoparticle size [13,14]. On the other side, the applied pressure with values more than 5 GPa with values from 8 to 10 GPa effectively fragment the W, Al_2_O_3_, and SiC hard reinforcement particles into UF and nanoparticle size, respectively [15,16,17]. Thus, HPT processing under pressure higher than 5 GPa (such as 10 GPa used in the present work) is required to obtain a complete fragmentation of reinforcement particles into UF and nanoparticle sizes. Therefore, the pressure can be considered the practical key that influences the fragmentation of the reinforcement in the HPT of the MMCs.

Applying the pressure in the HPT is followed by the lower die’s rotation, which continues up to the required number of revolutions is needed. Therefore, the fragmented SiC and Al_2_O_3_ particles can be homogeneously distributed in the Al matrix (Figure 7) relative to the agglomeration noted in the PM samples (Figure 5). Obtaining a homogenized distribution of the reinforcement was also observed after the HPT of the MMCs and intermetallic materials reinforced with micro-size particles [13,14,15,16,17,18,19,20,21]. Considering that agglomeration was one of the main defects of the ECAP and HPT processed Al MMCs reinforced with nano reinforcement particles [9,11,12,28]. Therefore, it was efficient to process the micro-sized reinforced Al MMCs samples by HPT under 10 GPa and 30 revolutions in producing a UFG Al matrix with a homogenized distribution of the fragmented SiC and Al_2_O_3_ particles.

### 3.3. Microhardness Results

The Microhardness distribution curves of PM and HPT Al samples can consider homogenized. A slight difference between the max and min microhardness values along PM and HPT Al samples diameter was noted (Figure 8a). The difference between the microhardness measurements of the PM and HPT Al samples did not exceed 2.5–5.7 Hv. Considering the strain increase from the center to the edge of the HPTed Al sample, the microhardness difference was higher in the HPT sample. The microhardness inhomogeneity index confirmed this observation. The microhardness inhomogeneity index of the PM Al sample, 7.2%, was lower than that of 8.7% of the HPTed sample (Figure 8b). However, the PM Al sample has a relatively higher homogeneity than the HPT sample. Nevertheless, due to the fine grain of the HPTed Al sample according to the Hall–Petche relationship (2) [31,32], it has a microhardness of 65.5 Hv, which is higher than that of 34.7 Hv of the PM Al sample by 88%.
(2)$$H=H_{o}\;+\;{\;K}_{H\;}d^{\frac{1}{2}}$$
where d, H_o_, and K_H_ are the grain size and constants. Furthermore, the dislocation density increase during the HPT processing increase microhardness H, similar to that of the strength increases according to the Taylor Equation (3) [33,34].
(3)$$\sigma =\sigma_{o\;}+{\;\alpha \mathrm{MGb}\rho }^{\frac{1}{2}}$$
where σ_o_ is the matrix material frictional stress, α is a constant, G is the shear modulus, b is the length of the Burgers vector of dislocation, M is the Taylor factor, and ρ is a dislocation density.

HPT Al samples’ average microhardness value was comparable to that ECAPed and HPTed Al powder, chip, and solid samples. HPTed Al powder microhardness was higher by 7–46.6% than Al solid and powder samples processed by ECAP and HPT [9,10,11,12,17,28,29]. However, the microhardness of the HPT Al powder was lower by 12.6–46.7% than that of the Al solid, powder, and chip processed by ECAP and HPT [10,17,29] due to the higher strain imposed. Interestingly, the higher value of the microhardness of the HPTed Al powder samples noted in previous work [10] cannot be explained logically, as the applied pressure and number of revolutions used were lower than those of the present and previous works [11,12,17,28,29].

The microhardness distribution patterns of the PM Al MMCs samples were similar to those of the PM Al samples. The microhardness varied above and under a datum with a value near the average microhardness of each sample. Moreover, the microhardness increased from the center to the edge in the HPTed one, as previously noted for different HPTed Al and Al alloys MMCs [9,10,11,12,14,17,28,29,35,36] (Figure 8a). The PM samples have a slightly higher degree of homogeneity than the HPTed one (Figure 8b). The inhomogeneity index of the PM Al-20% Al_2_O_3_ and SiC samples was lower by 14–29% than that of HPTed samples.

The difference between max and min microhardness values along the diameter of the PM and HPT Al-20% Al_2_O_3_ and SiC samples were in the range of 13.3–14.6 and 23.7–24.6 Hv, which is considered smaller than that of HPTed Al MMCs samples [11,12,28,35,36]. The difference between the microhardness in the center and the edge of the Al-30% nano Al_2_O_3_ [11,12], Al-5% CNT [28], Al chip-20% Al_2_O_3_ and SiC [35], and Al-20% nano Al_2_O_3_ [36] HPTed samples was 80–120, 45, 25–31, and 70 Hv, respectively. Therefore, the present results indicated that HPT under using 10 GPa and 30 revolutions effectively produces homogenous Al MMCs.

Although the HPT Al MMCs have a lower degree of homogeneity than the PM Al MMCs samples, the average microhardness of the HPTed Al MMCs was higher than that of the PM one. The average microhardness of HPTed Al-20% Al_2_O_3_ and SiC samples was higher by 31–42% than those of the PM samples. Interestingly, the HPTed Al-10% Al_2_O_3_ and SiC samples’ microhardness was higher by 9–12%% than that of PM Al-20% Al_2_O_3_ and SiC samples. According to the microstructure observations and the density results (Figure 3, Figure 4, Figure 5, Figure 6 and Figure 7), HPT samples have a high degree of densification, smaller grain, fine reinforcement particle sizes, and the homogenous distribution of the fragmented Al_2_O_3_ and SiC particles, which explains their high microhardness.

The deep comparison between the previous and current studies about Al MMCs fabricated using commercial or high-purity Al matrix using different processing methods indicates the following observations. The HPT of the Al-Al_2_O_3_ and SiC MMCs was effective over the PM method [25,26,37,38,39]. The microhardness of the Al-10% SiC [25]; Al-6% Al_2_O_3_ and Al-6% SiC [26]; Al-10% Al_2_O_3_ [37]; Al-5% A1N and Al-5%Ti [38]; and Al-1.5% Al_2_O_3_, Al-1.5% Al_2_O_3_ + 4%Al_4_C_3_, and Al-1.5% Al_2_O_3_ + 12%Al_4_C_3_ [39] was ranged from 130 to 150, 64 to 70, 130 to 150, 32.5 to 82.5, and 69 to 74 Hv, respectively.

Therefore, the HPT processing is successfully increasing the microhardness over that of Al MMCs processed by the PM combined with ball milling, hot pressing, and extrusion due to fine Al matrix grain and reinforcement particle sizes. Moreover, the increase in dislocation density and obstruction of dislocation motion by the fine reinforcement particles contribute to the microhardness increase. Additionally, the high pressure and imposed strain at room temperature contribute to producing fully dense samples with homogenous distributed UF reinforcement particles and overcome the effect of the high temperature used in the sintering, hot passing, and extrusion that leads to the grain growth and so microhardness decrease [25,26,37,38,39].

The comparison between the present and previous results can be extended to cover the precious results of the SPD processing of Al MMCs. The microhardness of the HPTed Al-20% Al_2_O_3_ noted in the present study can consider higher than the hardness of Al-Al_2_O_3_ ECAPed and HPTed samples [9,10,11,12,35,36]. The HPT Al-10 and 20% Al_2_O_3_ samples’ microhardness of 146 and 175.8 Hv were higher or near the microhardness of 158.6, 56–125, 125–250, 168.9, and 100 Hv of ECAP Al-10% nano Al-Al_2_O_3_ [9], HPTed Al-5% nano Al-Al_2_O_3_ [10], HPT Al-30% nano Al-Al_2_O_3_ [11,12], HPT Al chip-20% Al_2_O_3_ [35], and HPT Al-20% nano Al_2_O_3_ [36], respectively.

Although the HPTed Al-30% nano Al_2_O_3_ sample microhardness of 250 Hv [12] was higher than that of HPTed Al-20% micro Al_2_O_3_, it must consider the higher Vol% used in the previous work. However, the HPT processing of the Al-20% micro Al_2_O_3_ sample was more effective in fragmentation of the Al_2_O_3_ particles with a high degree of homogeneity, no trace of agglomeration and voids, noted in the SPD Al-5, 20, and 30% Al_2_O_3_ reinforced with nano-size Al_2_O_3_ particles [9,10,11,12].

Similar to that noted for the HPTed Al-Al_2_O_3_ samples, the HPTed Al-10 and 20% SiC samples’ microhardness was higher or near to those of different Al–SiC deformed by different SPD methods [6,7,35,40]. The microhardness of the HPTed Al-10 and 20% SiC of 163.8 and 207.2 Hv was higher by 70–21, 7, and 55% than those of ECAPed Al-10% SiC [6,7], HPTed Al chip-20% SiC [35], and roll bonding Al-SiC [40].

### 3.4. Tensile Properties

#### 3.4.1. The Tensile Strength and Elongation

The stress–strain curves of PM and HPT Al, Al-20% Al_2_O_3_, and SiC MMCs are indicated in Figure 9. The ultimate tensile strength (UTS) and proof strengths remarkably increased after the HPT relative to those of the PM. The PM samples UTS increased from 100.5, 359.1, and 388.7 MPa to 208.4, 462.2, and 533.5 MPa after the HPT of the Al, Al-20% Al_2_O_3_, and SiC samples, respectively. Therefore the UTS of the different samples was increased by 10–110% after the HPT. Furthermore, the HPT samples’ proof strength increased over those of the PM samples. Interestingly, the percentage increase in the HPTed samples’ strength tensile over those of the PM samples was observed to be near that of the increase in the HPTed samples’ microhardness over those of PM, which supports the results.

The formation of the Al MMCs effectively increases the tensile strength of the PM Al samples by 72–74% due to the secondary and geometrically necessary dislocation and thermal expansion dislocation strengthening mechanisms, respectively. Moreover, the HPTed Al MMCs tensile strength increase can be explained by the two mean reasons. First is the influence of the dislocations, including secondary and geometrically necessary dislocations. Second, the grain boundaries strengthen through HAGBs according to the Hall–Petche relationship and low-angle grain boundaries (LAGBs). The higher tensile strength of the HPTed Al MMCs over those of the PM samples is due to the higher dislocation density related to the SPD. Moreover, the effect, the HAGBs, and LAGBs contribute to further strengthening with the neglecting of Orowan mechanism strengthening.

The tensile strength increase after the SPD of different Al and Al alloy composites was the norm in most previous works [7,10,28,29]. The ECAP of Al-10% SiC and HPT of the Al-5% nano Al_2_O_3_, Al-CNT, and Al samples effectively increase the Al tensile strength by 58.3, 8.3, 36.2, and 52%. Interestingly, ECAP and HPT of the micro-size reinforced Al and Cu MMCs effectively improve the tensile strength over that of the nano-reinforced Al and Cu MMCs [7,16].

The elongation% of the PM Al, Al-20% Al_2_O_3_, and SiC samples decreased from 35.3, 21.47, and 13% down to 23.8, 19, and 11.9% after the HPT, respectively. The Al MMCs samples’ elongation% was smaller than the Al samples due to the obstruction of the dislocation motion by the reinforcement particles. These observations of the decreased tensile elongation of the Al and Al MMCs after HPT are congruent with those of different SPD Al MMCs [7,10,28]. The elongation% decreased due to Al’s grain refinement and UF reinforcement particles that further hamper the dislocation motion, as noted previously [16].

The elongation% decrease after the HPT of 32–37% was smaller than the tensile strength increase percentage. Therefore, the HPT of Al and Al MMCs powders can retain an acceptable level of tensile elongation%. This observation is due to the microstructure of the HPT samples into a combination of UFG and nano grains, as previously noted during Cu and Cu MMCs powder consolidation by HPT [16]. Moreover, high pressure is applied to enhance the density by decreasing the void%, so elongation% with an acceptable level can be conserved. Therefore, the HPT of Al and Al MMCs powders under 10 GPa and 30 revolutions at room temperature effectively conserve a reasonable level of elongation% with high tensile strength.

#### 3.4.2. Tensile Fracture Surfaces Morphology

The photomicrographs of the tensile fracture samples are shown in Figure 10. The PM and HPTed Al samples show a ductile fracture with the presence of dimples, as noted in (Figure 10a,b). The tensile fracture surface of the Al HPT sample (Figure 10b) has shallower and smaller dimples than the PM one because of grain refinement, as previously noted [29,41]. Moreover, smaller dimples in the HPTed Al samples are congruent with the elongation% decrease after the HPT. Furthermore, the higher degree of the reduction in a fracture sample area of the PM Al sample was noted more than that in the case of the HPT one, which confirmed the elongation results.

The Al MMCs fracture surface was covered by Al_2_O_3_ and SiC particles with small dimples, as noted in (Figure 10c–f). Interestingly, the Al_2_O_3_ and SiC particle sizes of the PM and HPT samples did not change and still have the same size noted in the microstructure observations. This observation clarifies that the tensile load did not contribute to any fracture of the reinforcement particles. The tensile fracture occurs along the Al matrix parallel to the interface between the Al matrix and the reinforcement.

The HPT samples’ tensile fracture surfaces were observed to be similar to that of the PM samples. However, smaller dimples and fine Al_2_O_3_ and SiC particles were noted in the HPTed samples (Figure 10c–f). The presence of fine Al_2_O_3_ and SiC particles hinders the dislocation motion and sizing of dimples. Moreover, the Al matrix grains refinement also decreases the dimple size [29,41]. The tensile fracture surface of the HPT samples shows a slight decrease in the ductile fracture features. These observations indicate the agreement between the tensile elongation% and tensile fracture surface morphologies analysis in the present work.

Furthermore, the fracture surface morphology indicates high bonding between reinforcement and matrix in the HPTed Al MMCs. The grain refinement and reinforcement fragmentation enhance the bonding, increasing the strength as the good bonding improves the ability of the load transfer from the matrix to the reinforcement particles. The good bonding in the case of the HPT Al MMCs samples was also confirmed through the density results and the low void% in the HPTed samples.

## 4. Conclusions

Through the current research, it was concluded the following:HPT processing produces fully dense Al and Al-MMCs samples with relative densities higher by 4% over the PM samples. The void% of 0.1, 0.7, and 0.3% of the HPTed Al, Al-20% Al_2_O_3_, and SiC samples, respectively, was lower by 85–96% than the PM samples.The HPT of Al MMCs with micro size matrix, reinforcement powders, and Vol% up to 20% effectively produce UFG Al MMCs samples.HPT of micro size Al and Al MMCs powder can refine the Al matrix grain size with a mixture of UFG and nanograins. The PM Al, Al-10% Al_2_O_3_, Al-20% Al_2_O_3_, Al-10% SiC, and Al-20% SiC samples Al matrix average grain size decreased by 99.5% down to 0.39, 0.3, 0.23, 0.28, and 0.2 µm, respectively, after the HPT.The Al MMCs matrix grain refinement combined with obvious fragmentation of the Al_2_O_3_ and SiC. The HPT processing contributes to the decrease of 99.5% in the particle sizes of Al_2_O_3_ and SiC with high compatibility and congruence between the results of the SEM and TEM.Although the HPT processing deteriorates the Al and the different Al MMCs samples’ deformation homogeneity, it increases their microhardness by 31–88% over those of the PM samples.The tensile strength results confirm the microhardness one, which indicates an increase of 10–110% in the tensile strength of the HPT samples over the PM samples. Furthermore, the tensile fracture surface results confirm the influence of the HPT on increasing the bonding between reinforcement and the Al matrix.

## Figures and Tables

**Figure 1 materials-15-08827-f001:**
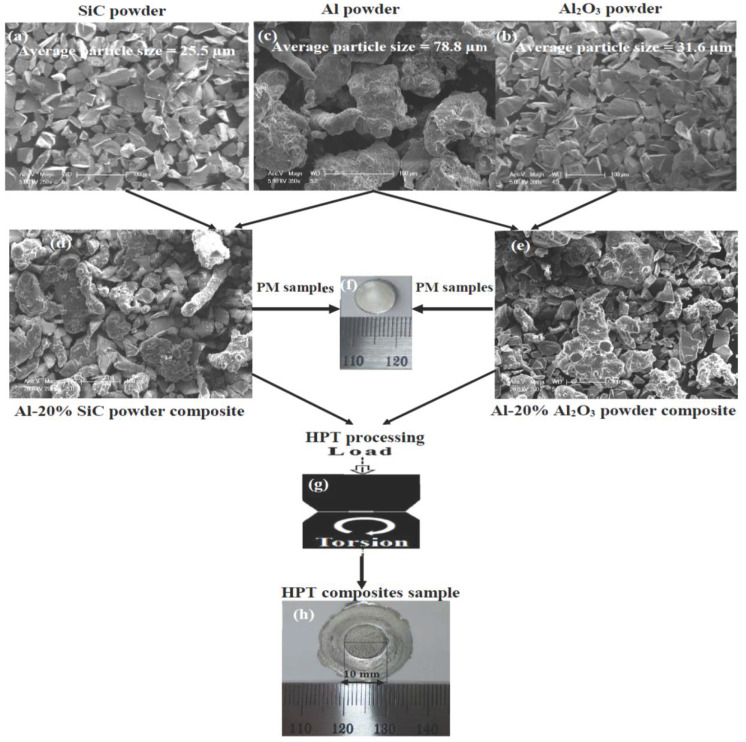
The PM and HPT processing diagram of Al-Al_2_O_3_ and SiC composites powder sample.

**Figure 2 materials-15-08827-f002:**
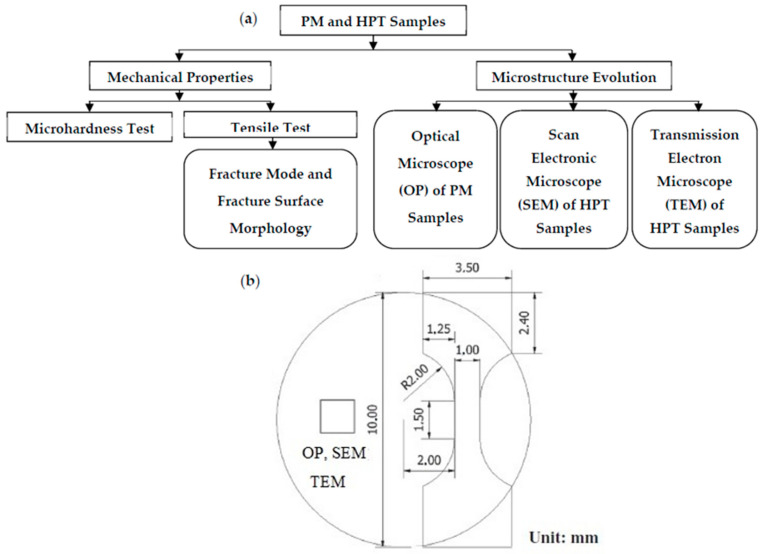
(**a**) Flow chart of the experimental testing; (**b**) the positions of the OP, SEM, and TEM observations; and tensile test sample.

**Figure 3 materials-15-08827-f003:**
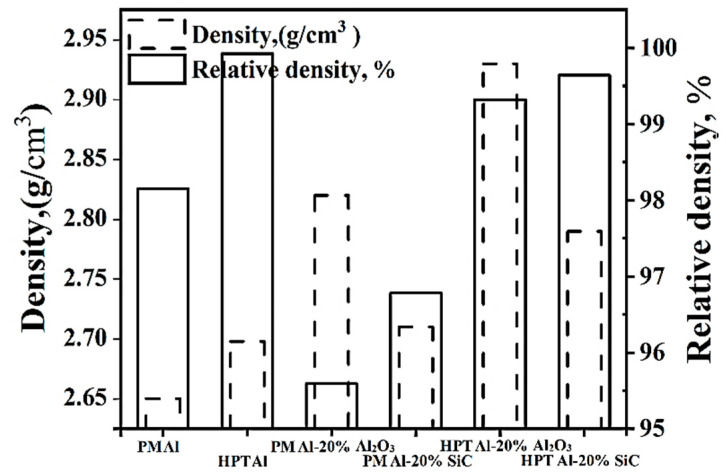
The Experimental and relative densities of the different samples.

**Figure 4 materials-15-08827-f004:**
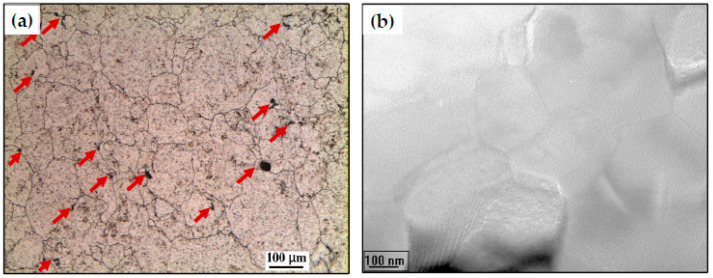
(**a**) Optical microscope micrograph of PM Al sample and (**b**) TEM micrograph bright field of HPT Al sample.

**Figure 5 materials-15-08827-f005:**
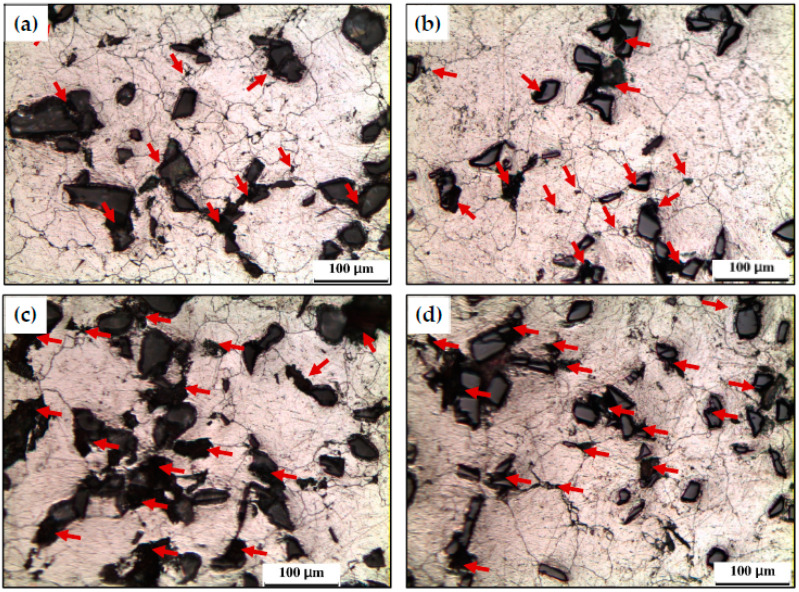
Optical microscope micrographs of PM (**a**) Al-10% Al_2_O_3_, (**b**) Al-10% SiC, (**c**) Al-20% Al_2_O_3_, and (**d**) Al-20% SiC samples.

**Figure 6 materials-15-08827-f006:**
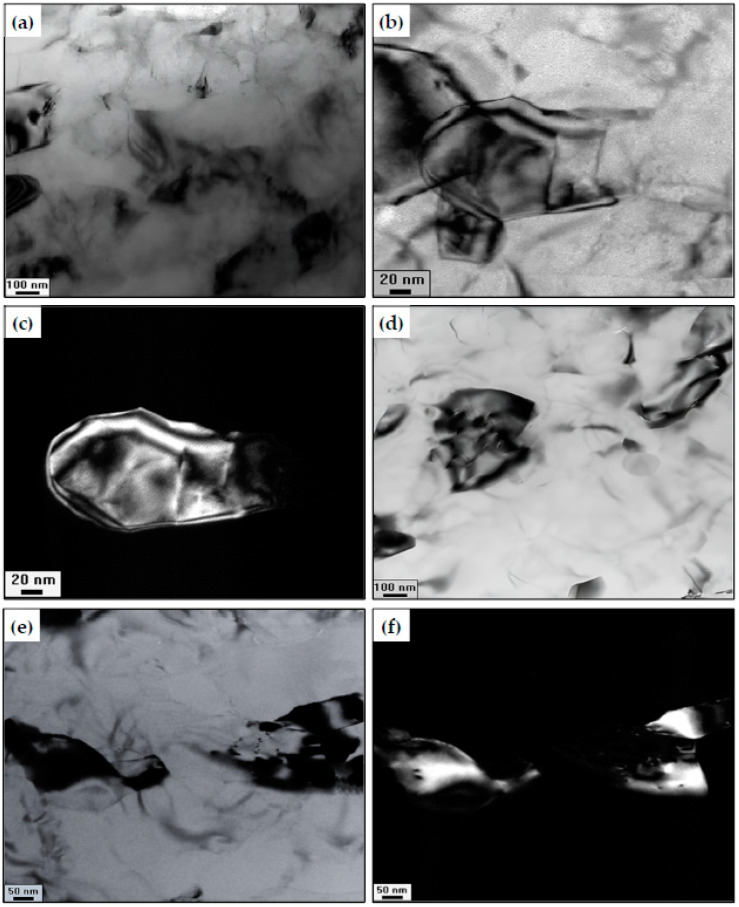
TEM micrographs of (**a**) bright field Al-10% SiC, (**b**,**c**) bright and dark fields Al-20% SiC, (**d**) bright field Al-10% Al_2_O_3_, and (**e**,**f**) bright and dark fields Al-20% Al_2_O_3_ samples after HPT processing.

**Figure 7 materials-15-08827-f007:**
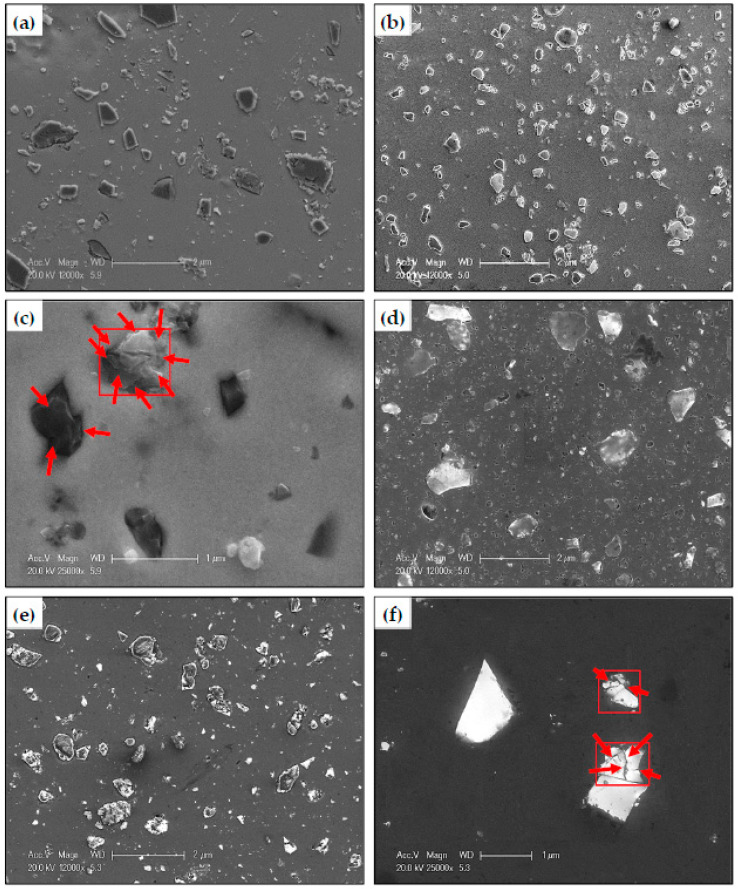
SEM micrographs of (**a**) Al-10% SiC, (**b**) Al-20% SiC, (**c**) the SiC particles fragmentation, (**d**) Al-10% Al_2_O_3_, (**e**) Al-20% Al_2_O_3_, and (**f**) the Al_2_O_3_ particles fragmentation after HPT.

**Figure 8 materials-15-08827-f008:**
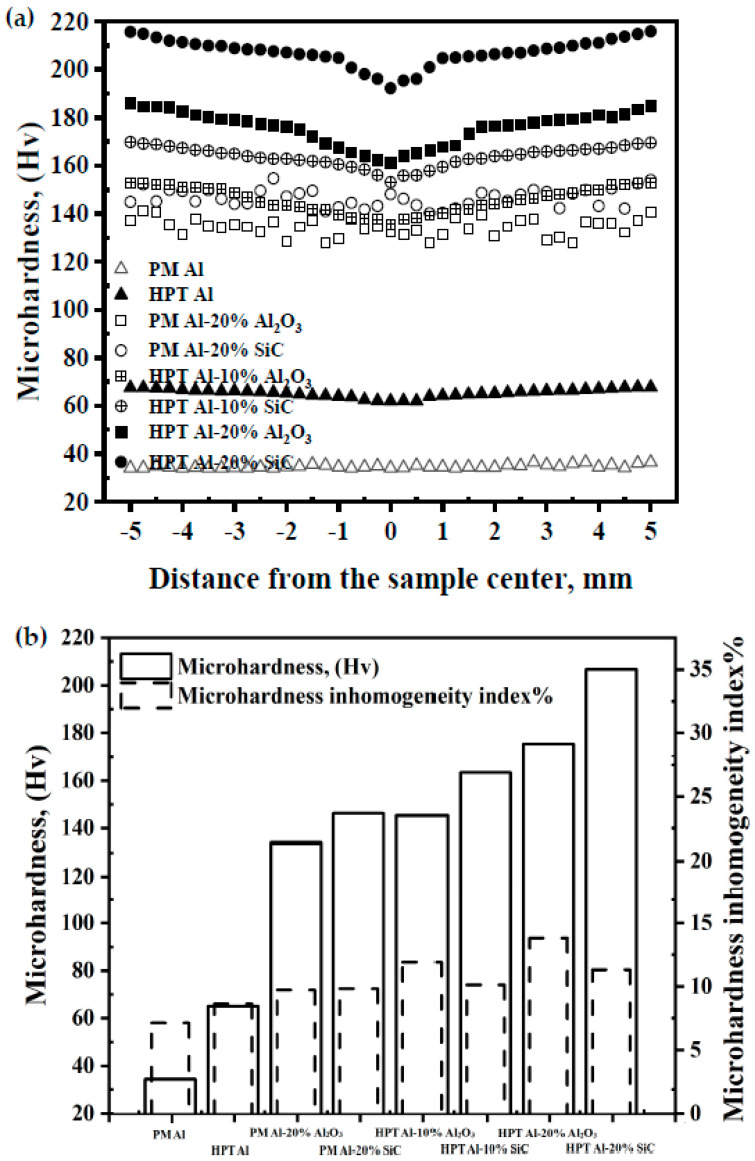
(**a**) Microhardness distribution across the diameter of the samples and (**b**) average microhardness and microhardness inhomogeneity index of the different samples.

**Figure 9 materials-15-08827-f009:**
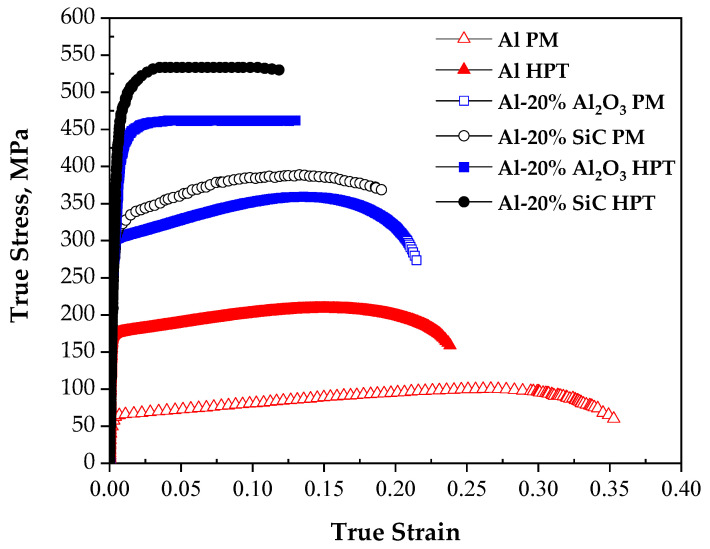
The true stress–true strain curves of the different samples.

**Figure 10 materials-15-08827-f010:**
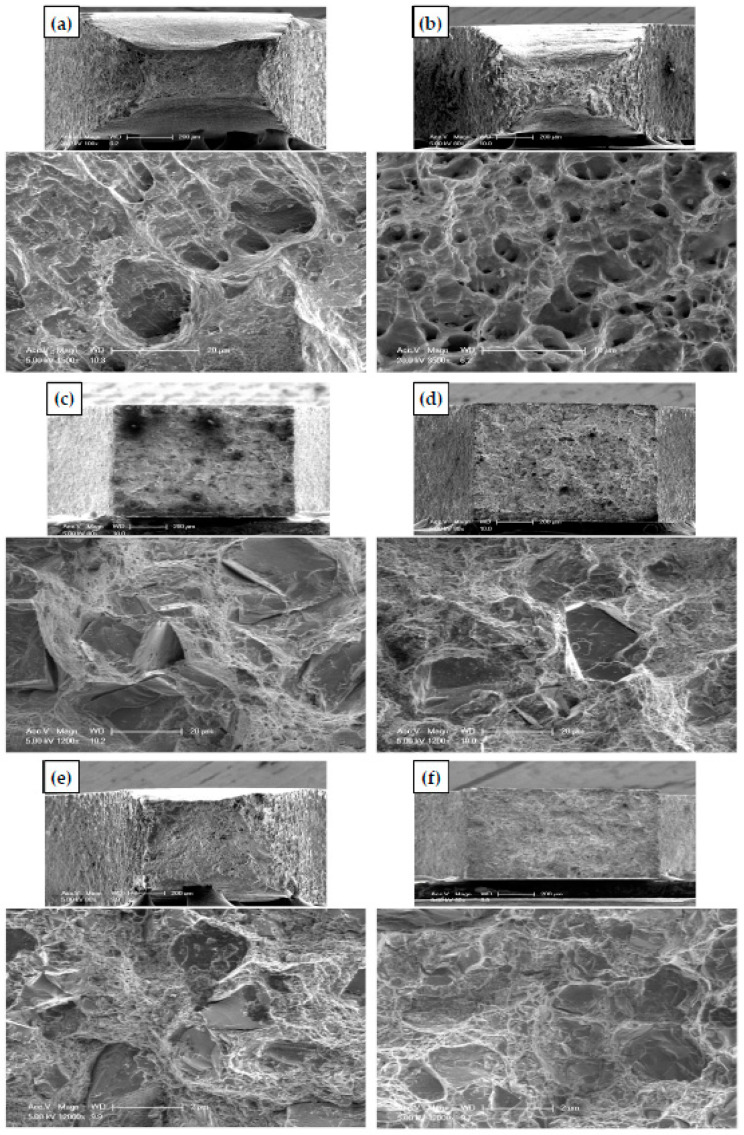
Fracture mode and surface morphology of (**a**) PM Al, (**b**) HPT Al, (**c**) PM Al-20% Al_2_O_3_, (**d**) PM-20% SiC, (**e**) HPT Al-20% Al_2_O_3_, and (**f**) HPT Al-20% SiC samples.

## Data Availability

The data presented in this study are available on request from the corresponding author. The data are not publicly available due to the extremely large size.

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
