# Peer review of "The Mechanical Properties of Aluminum Metal Matrix Composites Processed by High-Pressure Torsion and Powder Metallurgy"

_materials, 2022, doi:10.3390/ma15248827_

Round 1

Reviewer 1 Report

1. Abstract should be expanded sentences related to the results. The results of the study should be given as numerical percentages. 

2. Introductory sentence in abstract section is general statement please remove it.

3. What is novelty of this work and how it is different from pervious literature

4. Standards of test samples should be given.

5. Conclusions should be written in more detail adding numeric data. Also, give the results in items.

Author Response

AUTHOR REPLY TO THE REVEIWER COMMENTS ON materials-2083244

Dear the editor,

Thank you for conveying to me the recommendations of the referee.

We revised our manuscript altered in line with the suggestions of reviewer 1. We want to respond to his comments as follows.

Answer to the Reviewer#1

Abstract

(1)
Abstract should be expanded sentences related to the results. The results of the study should be given as numerical percentages. 

Answer:  The abstract has been rewritten to suit the reviewer's comments. Page no 1, Paragraph no 1, and  Lines15-23.

(2) Introductory sentence in abstract section is general statement please remove it.

Answer:  The sentence was removed.

Introduction

What is novelty of this work and how it is different from pervious literature?

Answer:  The novelty of the current work was emphasized and clarified. Page no 2, Paragraph no 6, and  Lines 82-88.

Materials and Methods

Standards of test samples should be given.

Answer: Preparing standard tensile test samples following the ASTM or ASM handbook volume 8, page 322   from The HPT with a 10 mm diameter was complex. Previous works concerned with the HPT-processed samples' tensile properties show the use of different dimensions with the need for special grips and Micro-tensile test machines [10, 16, 28, 29]. The shape and dimensions of the tensile sample relative to the original PM and HPT are indicated in figure 2(b).  Page 4.

Conclusions

Conclusions should be written in more detail adding numeric data. Also, give the results in items

 Answer: The conclusions were rewritten and modified.  Page no 15, Paragraph no 4, and Lines 467-489.

We appreciate the reviewer's and editor kind and instructive comments and hope that the revisions are satisfactory and that the revised version of the paper will be acceptable for publication in materials.

                                                                                                            Yours sincerely,

                                                                                                                 The authors

Reviewer 2 Report

The authors conducted a study related to production of aluminum matrix composites and its mechanical characterization. The high-pressure torsion was made for development of samples. It was reported that this process increases the micro hardness values and tensile strength. After all results were evaluated together, and they associated with each other, the manuscript can be accepted after these regulations.

- Title can be reconsidered.

-Abstract can be reorganized by demonstrating novelty, practical significance. Also it should be expanded sentences related to the experimental results. Some of the results of the study should be given as numerical percentages.

- Language of manuscript should be checked.

-Introduction part is tolerable. However, it should be expanded with more studies which support and explain to this subject.

* Effect of ball milling time on the structural characteristics and mechanical properties of nano-sized Y2O3 particle reinforced aluminum matrix composites produced by powder metallurgy route.

*Experimental study and analysis of machinability characteristics of metal matrix composites during drilling

*Mechanical properties and microstructure of composites produced by recycling metal chips

- The part of experimental design can be developed with some figures or flow charting.

- Conclusion part should be developed.

Author Response

AUTHOR REPLY TO THE REVEIWER COMMENTS ON materials-2083244

Dear the editor,

Thank you for conveying to me the recommendations of the referee.

We revised our manuscript altered in line with the suggestions of reviewer 2. We want to respond to his comments as follows.

Answer to the Reviewer#2

Title

(1) Title can be reconsidered.

Answer:  The title changed to "The Mechanical Properties of Aluminum Metal Matrix Composites Processed by High-Pressure Torsion and Powder Metallurgy" Page no 1, and  Lines 2-4.

Abstract

  • Abstract can be reorganized by demonstrating novelty, practical significance. Also it should be expanded sentences related to the experimental results. Some of the results of the study should be given as numerical percentages

Answer:  The abstract has been rewritten to suit the reviewer's comments. Page no 1 and Lines 15-23.

Language of manuscript

  • Language of manuscript should be checked.

Answer:  The English language was checked carefully throughout the manuscript.

Introduction

(1) Introduction part is tolerable. However, it should be expanded with more studies which support and explain to this subject.

* Effect of ball milling time on the structural characteristics and mechanical properties of nano-sized Y2O3 particle reinforced aluminum matrix composites produced by powder metallurgy route.

*Experimental study and analysis of machinability characteristics of metal matrix composites during drilling

*Mechanical properties and microstructure of composites produced by recycling metal chips

Answer:  The authors apologize for using three suggested manuscripts due to the following:

First, adding more information about those topics will increase the similarity in the manuscript.

Second, the three topics are away from the current study scope of interest, such as machinability characteristics.

Third, the first manuscript uses AA7075 and the other two use bronze; how to use these manuscripts in comparison with Al.

Fourth, the reviewer can kindly find that the effect of ball milling combined with HPT of Al-Al2O3 and the recycling of Al chip -Al2O3 and SiC composites was considered through references 12, 13, and 17 that can be considered enough.

Materials and Methods

(1) The part of experimental design can be developed with some figures or flow charting.

Answer: The Materials and Methods part was developed and supported by figure 2. A flow chart of the methods used to investigate the microstructure evolution and the mechanical testing was added in figure 2a. Moreover, a new figure indicating the positions of the OP, TEM, and SEM microscopes observations and tensile test samples was also added in figure 2b. Page no 4 .

Conclusion

(1) Conclusion part should be developed.

Answer: The conclusions were rewritten and modified.  Page no 15, Paragraph no 4, and Lines 467-489.

We appreciate the reviewer's and editor kind and instructive comments and hope that the revisions are satisfactory and that the revised version of the paper will be acceptable for publication in materials

                                                                                                            Yours sincerely,

                                                                                                                 The authors

Reviewer 3 Report

The use of the SPD methods for the fabrication of composites is an interesting direction in science that has received a new development in recent years. Although the authors’ paper is not new in a broad sense, it contains a lot of systematized and generalized data. However, the submitted paper cannot be published as is. The methodological part requires further clarification. There are questions about the statistical processing of the results. The results part should also be improved. There are many places where it is very difficult to follow the author's train of thought. In addition, the English of the manuscript should be improved. The main questions are listed below:

1. It should be clarified: have you used the HPT technique for processing a powder composite mixture or for a sintered solid composite? The first statement follows from the scheme in Figure 1, but the second statement follows from the text of the paper. Answer this question clearly and make appropriate corrections to the manuscript.

2. The powder sizes and scale marks in Fig. 1 are very hard to see. The size of the powders should be presented in the text of the manuscript, and the scale marks should be made clearer. The same applies to Figures 6 and 9.

3. Give the load used in the microhardness measurement.

4. Lines 144-145. ’… produced fully dens samples that can offer high mechanical properties because it is almost free of voids’. This is an incorrect statement. Firstly, there are still no results of measuring mechanical properties, and secondly, mechanical properties depend not only on the number of voids.

5. In many places in the manuscript it is difficult to follow the author's train of thought. Below I list a few examples:

Lines 178-179. It is very difficult to trace the correspondence between the material and the measured value. List all materials in sequence and then all relevant measured values.

Lines 224-225. Same

Lines 372-375. Same

Lines 284-285. Obviously, the authors mean the difference in the measured values ​​in each of the samples. However, one might also think that the authors are comparing PM and HPT samples.

6. Despite a lot of numerical data in the paper, the authors do not make any statistical processing of the results, but in most cases they limit themselves to the average value. This is a problem for a scientific article. Data scatter is always needed. In addition, the distribution of measured values ​​can be lognormal, which is often observed for the UFG structures. In this case, it is not correct to measure the average value at all.

7. Lines 272-273. ‘…are evenly distributed in the Al matrix’. What is meant by ‘evenly’? Uniformity in sample volume ? In this case, for which area of the disk-shaped HPT sample did the authors perform TEM analysis?

8. Due to the fact that the markers on some of the curves merge in Figure 8, it is difficult to trace the correspondence between the curve and the material.

9. Line 381. Why do you need to give two decimal places?

10. Lines 407-408. The authors give a different number of decimal places in the measured values.

11. English should be improved. This is due to both incorrect phrase constructions and the presence of typos. Below I list a few examples:

Lines 109, 222 and 465. ‘grain’ instead of ‘gain’

Lines 161-162. ‘The optical microscope photomicrographs of the PM samples microstructure …’. The phrase does not contain a verb.

Lines 344. ‘AlN’ instead of ‘AIN’

Author Response

AUTHOR REPLY TO THE REVEIWER COMMENTS ON materials-2083244

Dear the editor,

Thank you for conveying to me the recommendations of the referee.

We revised our manuscript altered in line with the suggestions of reviewer 3. We want to respond to his comments as follows.

Answer to the Reviewer#3

 Materials and Methods

(1) It should be clarified: have you used the HPT technique for processing a powder composite mixture or for a sintered solid composite? The first statement follows from the scheme in Figure 1, but the second statement follows from the text of the paper. Answer this question clearly and make appropriate corrections to the manuscript

Answer: The authors would like to indicate that figure 1 is clear, and it can see two sets of arrows. The first one comes from figures 1 (d) and (e), indicating the formation of PM samples that include cold compaction and then sintering. On the other hand, another set of arrows below the first one comes directly from the powders figures 1 (d) and (e) to HPT.

However, more clarification was performed by adding "cold compacted samples." Page no 3 , Paragraph no 1 , and  Line 104.

(2) The powder sizes and scale marks in Fig. 1 are very hard to see. The size of the powders should be presented in the text of the manuscript, and the scale marks should be made clearer. The same applies to Figures 6 and 9.

Answer: The size of the powders was added in the manuscript's text. "and particles size of 78.8, 31.6 and 25.5 µm". Page no 3, Paragraph no 1 , and  Lines 101-102.

Moreover, the scale marks in figure 1 are enhanced to be more apparent.

(3) Give the load used in the microhardness measurement.

Answer: The load used in the microhardness measurement was added to the manuscript. "used under an applied load of 100 gf and a dwell time of 15 s ".  Page no 4, Paragraph no 4, and  Line 132.

Results and Discussion

 (1) Lines 144-145. ’… produced fully dens samples that can offer high mechanical properties because it is almost free of voids’. This is an incorrect statement. Firstly, there are still no results of measuring mechanical properties, and secondly, mechanical properties depend not only on the number of voids.

Answer: The statement was corrected to" So, HPT of Al and Al MMCs produced fully dens samples, as it is almost free of voids." Page no 5, Paragraph no 4, and Lines 158-159.

(2)  In many places in the manuscript it is difficult to follow the author's train of thought. Below I list a few examples:

Answer: The statements were reconstructed to enables the reader to trace the data easily. More over other cases along the manuscript were check and it is ok.

  • Lines 178-179. It is very difficult to trace the correspondence between the material and the measured value. List all materials in sequence and then all relevant measured values.

Answer: The statement was corrected. Page no 7, Paragraph no 1, and  Lines 191-192.

  • Lines 224-225. Same

Answer: The statement was corrected. Page no 7, Paragraph no 6, and Lines 235-238.

  • Lines 372-375. Same

Answer: The statement was corrected. Page no 12, Paragraph no 6 , and  Lines 383-388.

  • Lines 284-285. Obviously, the authors mean the difference in the measured values ​​in each of the samples. However, one might also think that the authors are comparing PM and HPT samples.

Answer: The statement has nothing wrong to be corrected. But may be reviver means Lines 158-159. It was revised and corrected " So, HPT of Al and Al MMCs produced fully dens samples, as it is almost free of voids."

 The same comment was solved in the manuscript in the following places.

1- Page no 7, Paragraph no 3, and Lines 208-210.

2- Page no 9, Paragraph no 1, and Lines 254-257

3- Page no 12, Paragraph no 4, and Lines 372-376.

(3)  Despite a lot of numerical data in the paper, the authors do not make any statistical processing of the results, but in most cases they limit themselves to the average value. This is a problem for a scientific article. Data scatter is always needed. In addition, the distribution of measured values ​​can be lognormal, which is often observed for the UFG structures. In this case, it is not correct to measure the average value at all.

Answer: The average of the different microstructure parameters, such as grain size or particle size of the UFG materials, is the most popular used value. The reviewer asked kindly to check all used references in the manuscript or any other references he can select to see and to be sure of that. Therefore to compare the present result with the previous one, average values must be used. The average value in such material science work is important, as it can be further used in relations such as Hall-Petche. Moreover, if authors follow the suggestion to use the data scatter, what is the benefit, especially with the small size of the data samples obtained from the TEM or SEM? Furthermore, drawing the data scatter curve needs the variation between two parameters; if the grain size is the first one which one will be if we have only one variable, the processing method HPT and PM?

(4)  Lines 272-273. ‘…are evenly distributed in the Al matrix’. What is meant by ‘evenly’? Uniformity in sample volume ? In this case, for which area of the disk-shaped HPT sample did the authors perform TEM analysis?

Answer: The statement was changed to "can be homogeneously distributed in the Al matrix" to be more precise. Page no10, Paragraph no 2, and Lines 236.

The position of the TEM or other microstructure observation is indicated in figure 2(b) Page no 4 and mentioned in the manuscript. Page no 5, Paragraph no 2, and Lines140-141.

 (5)  Due to the fact that the markers on some of the curves merge in Figure 8, it is difficult to trace the correspondence between the curve and the material.

Answer: Figure 8 was modified with colors to be clearer so the reviewer can trace the correspondence between the curve and the material. Figure 8 is now named figure 9 in the new version. Page no 13.

(6)  Line 381. Why do you need to give two decimal places? & (7) Lines 407-408. The authors give a different number of decimal places in the measured values.

Answer: Along the manuscript, the numbers with only one decimal place were used. However, only the numbers related to the grain and particle size use numbers with two decimal places, as the approximation of them will change their values. For example, 0.06 µm equals 60 nm, and if approximated to 0.1 µm, which equals 100 nm, this will be wrong. All corrected numbers are highlighted throughout the manuscript.

Language of manuscript

English should be improved. This is due to both incorrect phrase constructions and the presence of typos. Below I list a few examples:

Lines 109, 222 and 465. ‘grain’ instead of ‘gain’

Lines 161-162. ‘The optical microscope photomicrographs of the PM samples microstructure …’. The phrase does not contain a verb.

Lines 344. ‘AlN’ instead of ‘AIN’

Answer:  The English language was checked carefully throughout the manuscript. The reviewer can check the correction of typing errors corrected and highlight on (page 6, line 175), (page 7, line 234), and (page 12, line 357). 

We appreciate the reviewer's and editor kind and instructive comments and hope that the revisions are satisfactory and that the revised version of the paper will be acceptable for publication in materials

                                                                                                            Yours sincerely,

                                                                                                                 The authors

Round 2

Reviewer 3 Report

Most of the requirements have been covered and the paper has been improved. I do not fully agree with the authors' ideas about the statistical processing of results, but I hope that it will be improved in their future works. Indeed, many authors use the average value, but in fact this is not always correct. As far as the TEM method is concerned, the TEM method is indeed very local. That is why it can be used very carefully to characterize the material. But now I won't talk about it. I have only one remark. Two phrases appear to be broken due to the deletion - see lines 58-61.

Author Response

AUTHOR REPLY TO THE REVEIWER COMMENTS ON materials-2083244 R2

Dear the editor,

Thank you for conveying to me the recommendations of the referee.

We revised our manuscript altered in line with the suggestions of reviewer 3. We want to respond to his comments as follows.

Answer to the Reviewer#3

Introduction

(1) Most of the requirements have been covered and the paper has been improved. I do not fully agree with the authors' ideas about the statistical processing of results, but I hope that it will be improved in their future works. Indeed, many authors use the average value, but in fact this is not always correct. As far as the TEM method is concerned, the TEM method is indeed very local. That is why it can be used very carefully to characterize the material. But now I won't talk about it. I have only one remark. Two phrases appear to be broken due to the deletion - see lines 58-61.

Answer: The authors would like to indicate their profound thanks to the reviewer and promise to be concerned about his valuable recommendation in their future work.

For the remark, the authors apologize for this mistake. The deletion was removed, and so the two phrases were corrected.  "The UFG and nanomaterials MMCs can be produced by consolidating UFG or nano-powders using down-to-up techniques. Unfortunately, this method includes many defects, such as inclusions, contamination, and a high percentage of voids." Page no 2, Paragraph no 2, and lines 58-61. 

We appreciate the reviewer's and editor kind and instructive comments and hope that the revisions are satisfactory and that the revised version of the paper will be acceptable for publication in materials

                                                                                                            Yours sincerely,

                                                                                                                 The authors
